# Electromyographic Analysis of Thigh Muscle Activity in Arthritic Knees During Sit-to-Stand and Stand-to-Sit Movements: Effects of Seat Height and Foot Position

**DOI:** 10.3390/healthcare13080920

**Published:** 2025-04-17

**Authors:** Hamad S. Al Amer, Mohamed A. Sabbahi, Hesham N. Alrowayeh, William J. Bryan, Sharon L. Olson

**Affiliations:** 1Department of Health Rehabilitation Sciences, Faculty of Applied Medical Sciences, University of Tabuk, Tabuk 71491, Saudi Arabia; 2School of Physical Therapy, Texas Woman’s University, 6700 Fannin Street, Houston, TX 77030, USA; msabbahi@twu.edu (M.A.S.); solson@twu.edu (S.L.O.); 3Physical Therapy Department, Faculty of Allied Health Sciences, Kuwait University, Sulaibekhat 90805, Kuwait; hrowayeh@hsc.edu.kw; 4Department of Orthopedics, The Methodist Hospital, 6565 Fannin Street, Houston, TX 77030, USA; wbryan@houstonmethodist.org

**Keywords:** electromyography, knee osteoarthritis, sit-to-stand, stand-to-sit, thigh muscles

## Abstract

**Background**: Knee osteoarthritis (OA) impairs functional mobility, including sit-to-stand and stand-to-sit movements. Thigh muscles stabilize the knee during these transitions, and variations in seat height and foot positioning may affect muscle activation. Assessing thigh muscle activity during these tasks may provide strategies to enhance function and guide targeted rehabilitation for individuals with knee OA. **Objective**: The aim of this study was to examine the EMG activity of the vastus medialis oblique (VMO), rectus femoris (RF), and biceps femoris (BF) muscles of arthritic knees during sit-to-stand and stand-to-sit movements when using varying seat heights and feet positions. **Methods**: The EMG activity was recorded from the three thigh muscles in the arthritic side during sit-to-stand and stand-to-sit movements under six different seating conditions from eight patients (three females; mean age: 64.6 ± 11.0 years). A three-way ANOVA was used to examine the effects of seat height, foot positioning, and movement type on muscle activation. **Results**: The results demonstrated significant interactions between muscle activation, movement type, and seating conditions (*p* = 0.022). The EMG activity of VMO and RF increased significantly during sit-to-stand movements from lower seat heights compared to knee-height seats (*p* < 0.05). RF activation was also significantly elevated during stand-to-sit transitions at low seat heights (*p* = 0.023). Additionally, sit-to-stand transitions with symmetrical foot placement elicited significantly greater VMO activation compared to BF activation (*p* < 0.05). While BF activation remained relatively low across most conditions, it was highest when the arthritic knee was positioned behind the sound foot during both movements. **Conclusions**: Seat height and foot positioning significantly impact thigh muscle activation in individuals with knee OA during sit-to-stand and stand-to-sit transitions. Lower seat heights require greater VMO and RF activation, indicating increased mechanical demands. Additionally, placing the arthritic knee behind the sound foot enhances BF activation, suggesting a potential strategy for targeted hamstring engagement. These findings provide directions for quadriceps and hamstring strengthening, alongside strategic seating adjustments to optimize functional mobility and reduce joint stress in individuals with knee OA.

## 1. Introduction

Knee osteoarthritis (OA) is a prevalent musculoskeletal condition significantly affecting functional mobility and quality of life [1,2]. It is characterized by the progressive degeneration of articular cartilage, leading to pain, stiffness, and impaired movement [3,4]. One of the most common functional challenges experienced by individuals with knee OA is difficulty transitioning from a seated to a standing position and vice versa [5,6,7,8]. The sit-to-stand and stand-to-sit movements are fundamental to daily living and require coordinated muscle activity to ensure stability and efficiency. Understanding the neuromuscular factors during these movements is critical for developing effective rehabilitation strategies.

Electromyography (EMG) has been widely used to assess muscle activity in individuals with knee OA [9,10,11]. Previous studies have demonstrated altered muscle activation patterns in these individuals, with compensatory strategies often observed to counteract pain and joint instability [12,13,14]. Thigh muscles are the key muscles involved in knee joint stabilization and mobility [15,16]. Variations in seat height and foot positioning can influence the demands placed on the thigh muscles, potentially affecting movement efficiency and joint loading [17,18,19]. Understanding how specific seating and foot positioning conditions impact muscle activation patterns in individuals with knee OA can provide valuable insights into optimizing functional performance and developing targeted rehabilitation interventions.

This study aimed to investigate the activity of the vastus medialis oblique (VMO), rectus femoris (RF), and biceps femoris (BF) muscles during sit-to-stand and stand-to-sit movements in individuals with arthritic knees under varying seat heights and foot positioning conditions. We hypothesized that variations in seat heights and foot positioning would influence the thigh muscles’ EMG activities due to altered mechanical demands during these movements.

## 2. Materials and Methods

### 2.1. Participants

The EMG activity was recorded from the thigh muscles of the arthritic knee of five male and five female participants, 7 to 14 days before having elective unilateral total knee arthroplasty (TKA). They were recruited using consecutive sampling using the following criteria: no other orthopedic disorders (lumbar, hip, or ankle pain) or neurological diseases (Parkinsonism, stroke, or head injury) which may limit function. None of the participants had undergone hip, knee, or spine surgery within the past 12 months. All participants reviewed and provided written informed consent approved by the Institutional Review Board of Texas Woman’s University (TWU) and The Methodist Hospital for the Protection of Human Subjects.

### 2.2. EMG Recording

EMG recording was performed using a Myosystem 1200 version 2.11 and a Telemyo 900 telemetry unit (Noraxon USA, Inc., Scottsdale, AZ, USA), with EMG signals sampled at 1000 Hz and with a sweep speed of 100 points/s. After cleaning the skin with alcohol, two surface electrodes (Ambu Blue Sensor M, Ambu, Inc., Ballerup, Denmark) were placed longitudinally over the mid-muscle belly of the vastus lateralis (VL), vastus medialis oblique (VMO), rectus femoris (RF), and biceps femoris (BF) in a bipolar configuration, maintaining a 2 cm inter-electrode distance. A single adhesive surface electrode was affixed over the head of fibula as a ground electrode to reduce background noise from the EMG recordings (Figure 1). The EMG activity of the VL was reported elsewhere [17]. Therefore, only VMO, RF, and BF activities are reported here. A single adhesive surface electrode was affixed over the head of fibula as a ground electrode to reduce background noise from the EMG recordings. Electrode placements were verified by observing the EMG activity during the performance of isometric contractions of the muscle. Then, the electrodes were secured on the skin using 3-M hypoallergic tape. The same certified clinical electrophysiologist performed all electrode placements for all participants. The captured EMG data were stored in a computer for off-line analysis.

### 2.3. Experimental Procedure

The experimental procedure required each participant to stand and return to a sitting position under six different conditions in the following orders: at knee-height with their feet together; at knee-height with their feet apart and the involved side’s foot in front (a semi-tandem heel-to-toe position with the arthritic side’s foot in front of the other); at knee-height with their feet apart and the normal side’s foot in front; at a low-height seat with their feet together (25% less than knee height); at a low-height seat with their feet apart and the involved side’s foot in front; and at low height with their feet apart and the normal side’s foot in front (Figure 2) (the tasks henceforward will be recognized as KHFT, KHVF, KHNF, LHFT, LHVF, and LHNF, respectively). In total, each participant completed 12 tasks (6 sit-to-stand tasks and 6 stand-to-sit tasks).

A backless, armless seat was used for the sit-to-stand and stand-to-sit movements. With both feet flat on the floor, thighs positioned at hip width, and arms crossed over the chest, the participants were asked to stand while evenly loading both feet in KHFT and LHFT. For the remaining conditions, participants were allowed to distribute their weight between both feet as needed to complete the task. After reaching a full standing position, they were instructed to return to a sitting position (i.e., stand-to-sit) while maintaining the same foot positions. While the EMG activity of the three muscles was recorded, each participant performed two trials of each task.

Afterward, the participants were seated on an electromechanical dynamometer (Biodex Inc., Shirley, NY, USA) to record EMG activity of the muscles while performing maximal voluntary isometric contractions (MVICs). The seat was adjusted for proper knee axis alignment, and the hip and knee were stabilized at 90° and 15° of flexion, respectively. The participants were instructed to maximally contract their quadriceps and then their hamstrings for 10 s each with 45 s of rest in between. A second trial was also performed after 2 min of rest. To ensure the MVICs were obtained, visual feedback of their peak force and verbal encouragement were provided and used.

### 2.4. Data Management

Raw EMG signals of the three muscles obtained during the experimental tasks and MVIC were analyzed using a Myosystem 1200 version 2.11 (Noraxon USA, Inc., Scottsdale, AZ, USA). After full-wave rectification and smoothing of raw signals with a 10 millisecond window, peak EMG signal amplitudes during the task trials and MVIC trials were obtained by analyzing the produced linear envelope. Because dynamic movements can sometimes produce greater EMG activation than MVIC [20], the functional task trials were also assessed to identify the maximum EMG signal. The highest EMG value recorded, whether during MVIC or functional task trials, was used for normalizing peak EMG signals. This normalization approach is commonly applied in EMG research [20,21,22] and has been recommended as a more precise method, as it ensures that all normalized peak EMG values remain at or below 100% of the maximum recorded signal [22]. Eventually, the peak EMG signals for three participants were normalized to the maximum signal recorded during the task trials, while for the remaining five participants, normalization was based on the maximum signal obtained during the MVIC. The outcome score for each muscle in each task was calculated as the average of the normalized peak EMG activity across the two trials.

### 2.5. Statistical Analysis

To examine the differences in the EMG activity between the thigh muscles during the two functional movements at different feet positions and seat heights, statistical analysis included three independent variables: (1) muscle with three levels (VMO, RF, and BF), (2) functional movement with two levels (sit-to-stand and stand-to-sit), and (3) condition with six levels (KHFT, KHVF, KHNF, LHFT, LHVF, and LHNF). The outcome was the normalized peak EMG activity measured in percent. Using SPSS for Windows version 25.0 (IBM, Armok, NY, USA), a 3 × 2 × 6 analysis of variance (ANOVA) for repeated measures was used to analyze the data. When statistically significant differences were found among the levels of the independent variables, cell mean comparisons were conducted using Bonferroni post hoc tests to determine the differences. The level of significance was set at alpha 0.05.

## 3. Results

### 3.1. Participants

Two female participants could not complete some of the tasks. One participant was unable to stand from LHNF, and the other one from the LHFT, LHVF, and LHNF. Therefore, their records were not used in the final analysis. The anthropometric data of the remaining eight participants are presented in Table 1.

### 3.2. EMG Activity

The means and standard deviations of the normalized EMG for the thigh muscles in each functional movement and condition are shown in Table 2.

The three-way ANOVA showed significant interaction between the three independent variables (muscle, functional movement, and condition), *p =* 0.022, (Table 3). In this case, follow-up analysis was conducted using two-way ANOVAs, main effect tests, and pairwise comparisons, as described in the following sections.

### 3.3. EMG Activity of VMO During Sit-to-Stand and Stand-to-Sit with Varied Seat Height and Feet Positions

Two-way ANOVA showed significant interaction in the VMO EMG activity between sit-to-stand and stand-to-sit at the varied seat height and feet positions, *F* (5, 35) = 7.88, *p* < 0.0005, (Figure 3a).

The EMG activity of the VMO varied during the sit-to-stand movement, ranging from 32.5% at KHVF to 77.9% at LHNF. However, main effect tests revealed no significant differences in EMG activity across these conditions (*p* = 0.109). In contrast, during the stand-to-sit movement, VMO activity remained more stable, with a narrower range of 29.24% at KHVF to 44.8% at LHNF, and similarly, no significant differences were observed between conditions (*p* = 0.109). Overall, VMO EMG activity was significantly higher in the sit-to-stand movement than in stand-to-sit under the LHFT (*p* = 0.035), LHVF (*p* = 0.016), and LHNF (*p* = 0.011) conditions (Table 4).

### 3.4. EMG Activity of RF During Sit-to-Stand and Stand-to-Sit with Varied Seat Height and Feet Positions

Two-way ANOVA showed significant interaction in the RF EMG activity between sit-to-stand and stand-to-sit at the varied seat height and feet positions; *F* (5, 35) = 4.51, *p* = 0.003, (Figure 3b).

Main effect analysis results for RF EMG activity are presented in Table 5. During the sit-to-stand movement, RF EMG activity gradually increased from 30–31% at KHFT and KHVF to 61.1% at LHNF, although this increase was not statistically significant (*p* = 0.156). In contrast, significant differences were observed across conditions during the stand-to-sit movement (*p* = 0.023). Pairwise comparisons revealed a significant increase in RF EMG activity from 24.10% at KHVF to 33.89% at LHFT (*p* = 0.001), 34.41% at LHVF (*p* = 0.043), and 37.42% at LHNF (*p* = 0.030) (Table 6). Additionally, RF EMG activity was significantly higher in sit-to-stand than in stand-to-sit under the LHFT (53.4% vs. 33.8%; *p* = 0.016), LHVF (60.7% vs. 34.4%; *p* = 0.008), and LHNF (61.1% vs. 37.4%; *p* = 0.047) conditions.

### 3.5. EMG Activity of BF During Sit-to-Stand and Stand-to-Sit with Varied Seat Height and Feet Positions

Figure 3c illustrates the EMG activity of the BF during sit-to-stand and stand-to-sit movements across various seat heights and foot positions. BF activation increased from 19.3% at KHFT to 23.3% at KHVF, peaking at 40.9% at KHNF in both movements, before decreasing to a range of 30.2–32.2% at LHFT, LHVF, and LHNF. However, two-way ANOVA showed no significant interaction between movement type and condition, *F* (5, 35) = 1.75, *p* = 0.148. Additionally, neither the main effect of movement, *F* (1, 7) = 0.79, *p* = 0.403, nor condition, *F* (5, 35) = 2.14, *p* = 0.083, reached statistical significance. As a result, no further analysis was conducted for the BF.

### 3.6. EMG Activity of Thigh Muscles During Sit-to-Stand Movement with Varied Seat Height and Feet Positions

Two-way ANOVA showed significant interaction between the thigh muscles’ EMG activity during sit-to-stand movement at the varied functional conditions, *F* (10, 70) = 2.91, *p* = 0.004, (Figure 4a).

The main effects analysis for thigh muscle EMG activity is presented in Table 7. At KHFT, the VMO exhibited the highest EMG activity (42.4%), while the BF showed the lowest (19.3%). Across other knee-height movement conditions, muscle activation remained comparable, ranging from 23.3% to 32.5% at KHVF and 40.9% to 42.5% at KHNF. In the low-height condition, BF activity remained consistently low (30.2–32.2%), whereas RF activity increased from 53.4% at LHFT to approximately 61% at LHVF and LHNF. The VMO showed a sharp increase to 65.5% at LHFT and then slightly decreased to 56.4% at LHVF before rising again to 77.9% at LHNF. However, the main effects analysis revealed significant differences in thigh muscle activity only at KHFT (*p* = 0.022) and LHFT (*p* = 0.041). Pairwise comparisons further indicated that VMO EMG activity was significantly higher than BF during rising from KHFT (*p* = 0.004) and LHFT (*p* = 0.008) (Table 8).

### 3.7. EMG Activity of Thigh Muscles During Stand-to-Sit Movement with Varied Seat Height and Feet Positions

Figure 4b illustrates the EMG activity of the thigh muscles during the stand-to-sit movement across different seat heights and foot positions. During KHFT, the BF exhibited the lowest EMG activity (19.2%) compared to the RF (35.7%) and VMO (38.1%). However, at KHNF, the BF showed the highest activation (47.2%), surpassing the RF (31.9%) and VMO (38.4%). Overall, no notable differences were observed among the three thigh muscles under the remaining testing conditions. The two-way ANOVA revealed no significant interaction between thigh muscle EMG activity and the varied functional conditions; *F* (10, 70) = 1.79, *p* = 0.077. Consequently, no follow-up analysis was carried out.

## 4. Discussion

The present study investigated the EMG activity of the VMO, RF, and BF muscles in individuals with arthritic knees during sit-to-stand and stand-to-sit movements across varying seat heights and feet positions. The findings revealed a significant interaction effect of both seat height and feet positioning on muscle activation patterns. Specifically, the VMO and RF showed consistent significant increase in the EMG activity during sit-to-stand movement from low seat height regardless of the feet position. Second, the RF muscle showed a significant increase during stand-to-sit movement at low seat height, regardless of foot position, in comparison to normal seat height with placing the involved seat anteriorly. Third, placing both feet at the same level during sit-to-stand movement, regardless of seat height, significantly increased the VMO activity in comparison to the BF. Overall, and when comparing the three thigh muscles, the VMO demonstrated the highest EMG activity, particularly during sit-to-stand transitions at low seat heights. In contrast, the BF exhibited the least activation across most conditions, except during the stand-to-sit movement under the KHNF condition, where it showed the highest activity level, although this difference was not statistically significant.

The significant increase in VMO and RF activation with lower seat heights indicates the requirement of greater effort to rise from a seated position. Higher chairs require less vertical force and reduce knee joint and muscle forces, making it easier to stand up. On the other hand, lower chairs increase the difficulty due to greater vertical force requirements and more significant forward displacement of the center of gravity (COG) [18]. A similar pattern was observed in the VL, with higher EMG activity when participants rose from a low-height seat compared to a knee-height seat [17]. The findings of this study support the concept that individuals with knee OA may experience greater difficulty and potential discomfort when rising from lower seating positions. This is particularly relevant in everyday scenarios where seating adjustments are not always feasible.

Additionally, the results of this study showed that the VMO was consistently significantly activated in comparison to BF during sit-to-stand movement at both low and knee seat heights when placing both feet together. This suggests that VMO plays a dominant role in facilitating sit-to-stand transitions when the feet are symmetrically positioned, emphasizing its importance in knee extension mechanics. Although previous research demonstrated increased co-contraction between the VL and BF in patients with knee OA during sit-to-stand movements [23], the significantly lower BF activation observed in this study suggests that knee extension is primarily driven by the quadriceps muscle. Roldán-Jiménez et al. [24] examined muscular activity fatigue during repeated sit-to-stand tests in healthy adults. They found that the VMO showed the highest activation and fatigue levels, emphasizing its critical role in performing the STS movement. Conversely, the BF showed varied activity levels with increased repetitions, indicating its supportive and stabilizing role in the sit-to-stand movement by aiding knee flexion and hip extension. These findings are relevant in the context of rehabilitation, as they suggest the necessity of targeting quadriceps strength in interventions aimed at improving sit-to-stand performance in individuals with knee OA.

Descending to a lower seat imposes greater demands on the knee extensors, likely due to the increased eccentric control required to stabilize the knee joint and regulate movement velocity. This challenge is particularly pronounced in individuals with knee OA, where muscle weakness and joint instability may further compromise joint movement [25]. In this study, the RF muscle exhibited a significant increase in activation during the stand-to-sit movement at low seat height, regardless of the feet position. This finding suggests the crucial role of the RF muscle, as a biarticular muscle, in controlling deceleration and maintaining safer movement coordination during the transition to a seated position, particularly in clinical populations with compromised knee function.

Previous studies have emphasized the importance of hamstring strength as well in knee OA management [26,27]. Aslan et al. [26] reported that individuals with knee OA exhibit weakness in both quadriceps and hamstring muscle groups. Al-Johani et al. [27] found that strengthening the hamstrings, in addition to the quadriceps, significantly improved knee pain, increased range of motion, and reduced functional limitations in patients with knee OA. Our results showed that sitting down on a knee-height seat with the foot of the arthritic knee positioned behind the sound foot (KHNF) imposed the highest demand on the BF muscle. The same was observed when standing up from this position. Further lowering of the seat height did not increase BF activation. These findings suggest that the BF muscle would benefit most from sit-to-stand and stand-to-sit exercises in the KHNF position, with no additional adjustments in foot placement or seat height necessary to further enhance activation.

## 5. Clinical Implications

Previous research concluded that patients post-TKA tend to rely on their uninvolved leg during functional tasks like sit-to-stand, potentially accelerating OA in that leg. Therefore, bilateral exercises may not sufficiently strengthen the affected leg, so adding unilateral exercises is advised to enhance quadriceps strength and prevent compensation [20]. This approach might also benefit patients with unilateral knee OA. Furthermore, it has been shown that sit-to-stand-based knee extensors exercise programs can effectively enhance muscle strength and reduce the muscular demands of daily activities in physically frail elders [28]. In fact, quadriceps muscle strength was found to be a key predictor of independence among the older population [29]. Additionally, strengthening the hamstring alongside the quadriceps was shown to further enhance rehabilitation outcomes in knee OA [27]. The results observed in this study could provide guidance for an effective rehabilitation program for individuals with knee OA. Training protocols that incorporate targeted foot positioning can optimize muscle strengthening exercises. The increased activation of the VMO and RF during sit-to-stand tasks suggests that focused strengthening of these muscles may enhance functional independence and performance. Additionally, the higher BF activation in the KHNF position indicates that sit-to-stand and stand-to-sit exercises in this posture may be particularly effective for targeting BF engagement without requiring further adjustments to foot placement or seat height.

In terms of environmental modifications for patients with knee OA, the increased quadriceps demand at lower seat heights suggests that occupational therapists should consider adjusting seat height to ease functional transitions. By optimizing environmental factors and educating patients on proper positioning, mobility and quality of life may be enhanced.

## 6. Limitations

This study has some limitations that should be recognized. First, the relatively small sample size may restrict the generalizability of the findings. Additionally, the limited sample size and imbalance between male and female participants (five males and three females) did not allow for sex-based subgroup analyses. Second, the study only included participants scheduled for unilateral TKA, which may not represent the broader population of individuals with varying severity of knee OA. Future research should consider larger sample sizes, explore potential sex differences in EMG activation among individuals with knee OA, and include participants with varying degrees of disease severity to enhance generalizability. Third, this study relied solely on EMG recordings. Future studies should incorporate additional biomechanical tools, such as motion analysis systems and force plates, to provide a more comprehensive assessment of movement patterns. Fourth, the EMG analysis in this study was limited to peak amplitude without assessing activation timing or muscle coordination patterns. While peak EMG amplitude is a commonly used parameter to reflect muscle demands, future studies should consider time-domain analyses to explore muscle synchronization and activation sequencing. Fifth, the EMG recordings were not segmented into specific movement phases (e.g., early vs. late phases of the task), limiting our ability to assess phase-specific muscle activity. Future research is encouraged to incorporate phase-specific analyses for a deeper understanding of muscle activation strategies. Sixth, motor unit recruitment frequency characteristics, such as mean frequency from a Fourier transform, were not analyzed. These analyses could provide valuable insights into neuromuscular control and are recommended for future research.

## 7. Conclusions

In conclusion, this study demonstrated that lower seat height and feet positioning significantly influence EMG activity in the thigh muscles during sit-to-stand and stand-to-sit movements in individuals with arthritic knees. The increased activity of the VMO and RF, especially during the transition from a low seat, suggests that greater knee extensor effort is required when rising from lower surfaces, whereas the comparatively lower activation of the BF suggests a reliance on quadriceps-driven movement. However, the higher BF activation observed in the KHNF position suggests that this foot placement may be particularly effective in engaging the BF muscle. These findings support the clinical importance of targeted quadriceps strengthening in rehabilitation programs for patients with knee OA. Moreover, the study emphasizes the potential benefit of modifying environmental factors, such as seat height, to ease functional mobility and reduce joint stress in individuals with knee OA.

## Figures and Tables

**Figure 1 healthcare-13-00920-f001:**
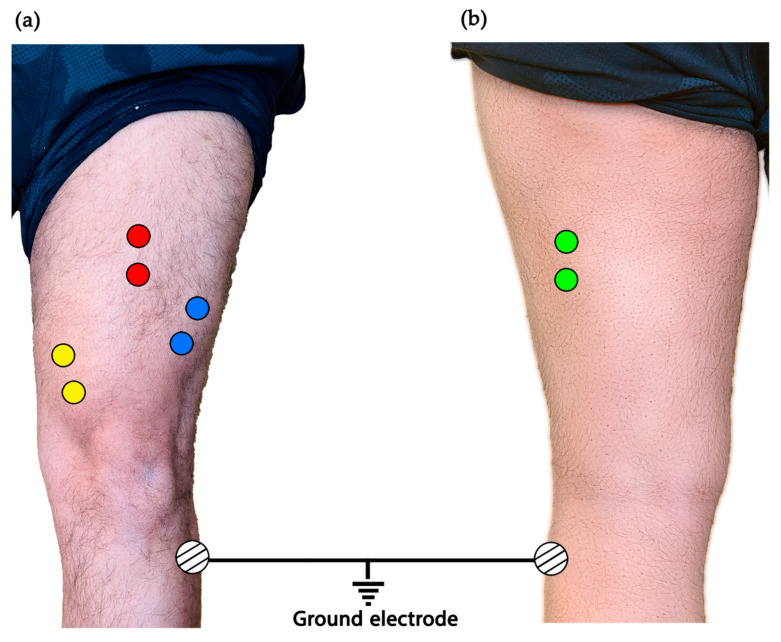
Electrode placement for recording EMG activity of the left thigh muscles. (**a**) Anterior view: vastus lateralis (blue circles), rectus femoris (red circles), and vastus medialis oblique (yellow circles). (**b**) Posterior view: biceps femoris (green circles).

**Figure 2 healthcare-13-00920-f002:**
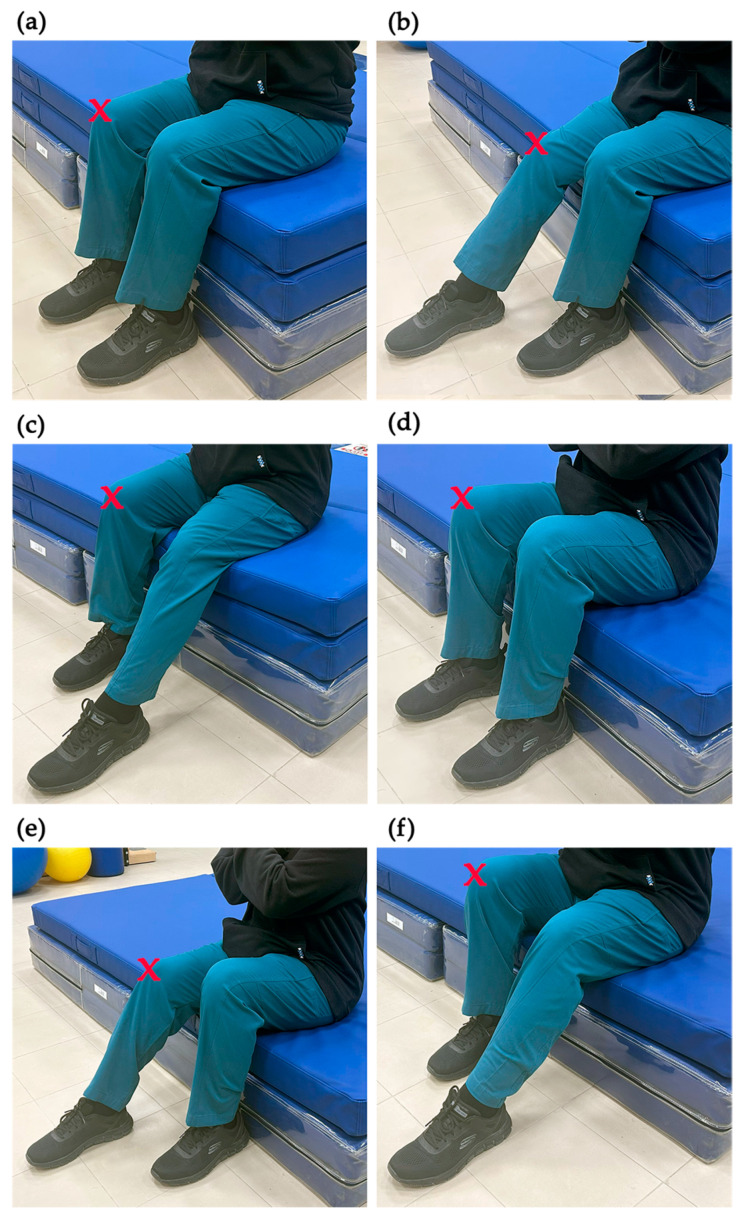
The six testing conditions for sit-to-stand and stand-to-sit movements. (**a**) At knee height with their feet together (KHFT). (**b**) At knee height with the involved front (KHVF). (**c**) At knee height with the normal front (KHNF). (**d**) At low height with their feet together (LHFT). (**e**) At low height with the involved front (LHVF). (**f**) At low height with the normal front (LHNF). Note: The (X) symbol denotes the arthritic knee.

**Figure 3 healthcare-13-00920-f003:**
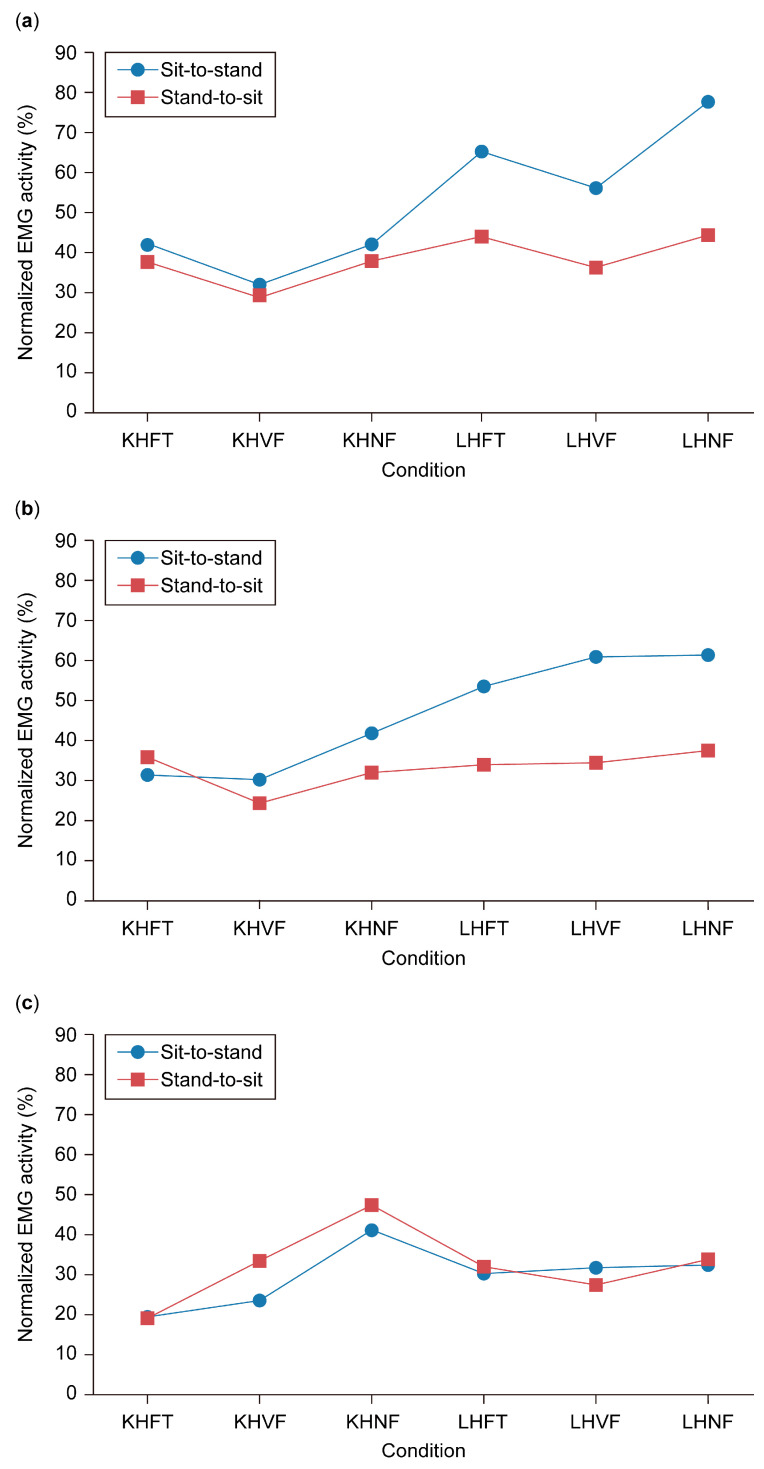
Electromyographic activity during sit-to-stand and stand-to-sit at different seat heights and foot positions. (**a**) Vastus medialis oblique (VMO). (**b**) Rectus femoris (RF). (**c**) Biceps femoris (BF). KHFT, knee height feet together; KHVF, knee height involved front; KHNF, knee height normal front; LHFT, low height feet together; LHVF, low height involved front; LHNF, low height normal front.

**Figure 4 healthcare-13-00920-f004:**
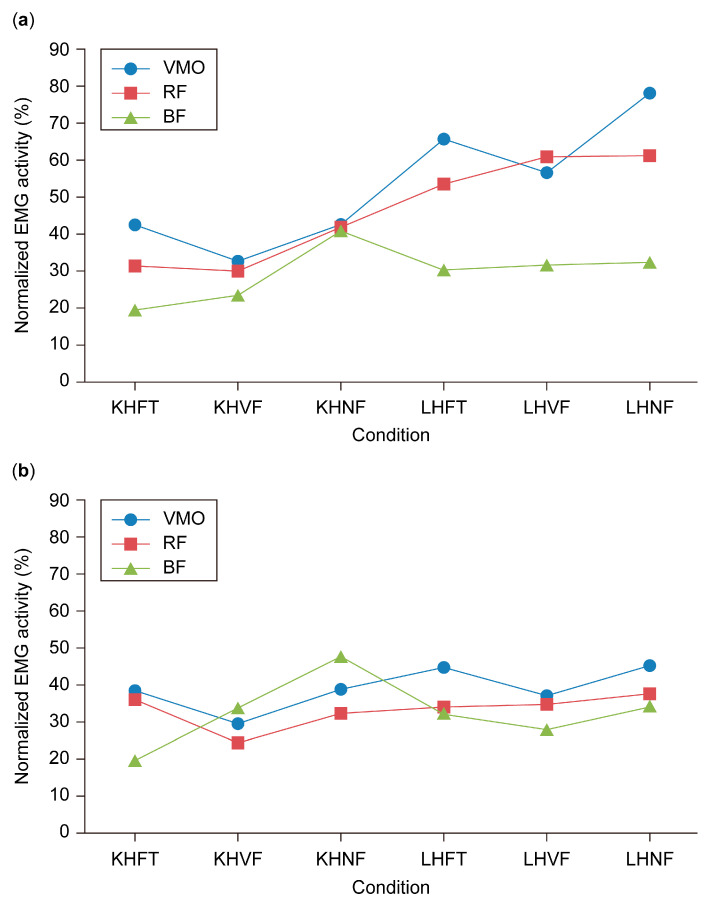
Electromyographic activity of the thigh muscles at different seat heights and foot positions. (**a**) Sit-to-stand. (**b**) Stand-to-sit. KHFT, knee height feet together; KHVF, knee height involved front; KHNF, knee height normal front; LHFT, low height feet together; LHVF, low height involved front; LHNF, low height normal front.

**Table 1 healthcare-13-00920-t001:** Anthropometric characteristics of study participants (*n* = 8).

Characteristic	Mean	SD	Range
Age (years)	64.6	11.0	44.8–76.2
Height (m)	1.72	0.09	1.6–1.8
Weight (kg)	92	24.8	55–130
Body mass index (kg/m^2^)	34.0	8.8	24.0–46.5

SD, standard deviation; m, meter; kg, kilogram.

**Table 2 healthcare-13-00920-t002:** Thigh muscle activity (%) across testing conditions.

Condition	Sit-to-Stand						Stand-to-Sit					
	VMO		RF		BF		VMO		RF		BF	
	M	SD	M	SD	M	SD	M	SD	M	SD	M	SD
KHFT	42.44	18.29	31.39	14.66	19.35	16.53	38.19	13.97	35.70	27.34	19.26	10.60
KHVF	32.53	14.60	29.96	17.46	23.37	17.96	29.24	12.13	24.10	14.01	33.40	26.68
KHNF	42.56	18.74	41.78	24.51	40.93	37.74	38.42	16.70	31.95	18.38	47.24	27.74
LHFT	65.58	17.60	53.43	27.23	30.28	33.33	44.39	24.70	33.89	15.94	32.01	35.03
LHVF	56.45	22.14	60.76	35.91	31.53	30.42	36.82	11.06	34.41	18.55	27.59	28.56
LHNF	77.90	27.54	61.14	29.81	32.23	35.29	44.86	12.93	37.42	22.51	33.88	34.72

VMO, vastus medialis oblique; RF, rectus femoris; BF, biceps femoris; M, mean; SD, standard deviation; KHFT, knee height feet together; KHVF, knee height involved front; KHNF, knee height normal front; LHFT, low height feet together; LHVF, low height involved front; LHNF, low height normal front.

**Table 3 healthcare-13-00920-t003:** Results of three-way ANOVA for EMG activity.

Factor	*df*	*F*	*p*-Value
Muscle	2, 14	1.88	0.189
Movement	1, 7	10.93	0.013 *
Condition	5, 35	5.50	0.001 *
Muscle × movement	2, 14	3.37	0.064
Muscle × condition	10, 70	2.42	0.015 *
Movement × condition	5, 35	9.51	<0.0005 *
Muscle × movement × condition	10, 70	2.28	0.022 *

* Significant at α = 0.05.

**Table 4 healthcare-13-00920-t004:** Main effect test results for vastus medialis oblique EMG activity.

Main Effect	*df*	*F*	*p*-Value
Sit-to-stand across all conditions	5, 3	4.95	0.109
Stand-to-sit across all conditions	5, 3	2.39	0.252
Sit-to-stand vs. stand-to-sit at KHFT	1, 7	2.576	0.153
Sit-to-stand vs. stand-to-sit at KHVF	1, 7	0.970	0.357
Sit-to-stand vs. stand-to-sit at KHNF	1, 7	0.962	0.359
Sit-to-stand vs. stand-to-sit at LHFT	1, 7	6.81	0.035 *
Sit-to-stand vs. stand-to-sit at LHVF	1, 7	9.84	0.016 *
Sit-to-stand vs. stand-to-sit at LHNF	1, 7	11.91	0.011 *

KHFT, knee height feet together; KHVF, knee height involved front; KHNF, knee height normal front; LHFT, low height feet together; LHVF, low height involved front; LHNF, low height normal front. * Significant at α = 0.05.

**Table 5 healthcare-13-00920-t005:** Main effect test results for rectus femoris EMG activity.

Main Effect	*df*	*F*	*p*-Value
Sit-to-stand across all conditions	5, 3	3.67	0.156
Stand-to-sit across all conditions	5, 3	16.03	0.023 *
Sit-to-stand vs. stand-to-sit at KHFT	1, 7	0.15	0.708
Sit-to-stand vs. stand-to-sit at KHVF	1, 7	1.29	0.292
Sit-to-stand vs. stand-to-sit at KHNF	1, 7	1.42	0.272
Sit-to-stand vs. stand-to-sit at LHFT	1, 7	10.0	0.016 *
Sit-to-stand vs. stand-to-sit at LHVF	1, 7	13.3	0.008 *
Sit-to-stand vs. stand-to-sit at LHNF	1, 7	5.79	0.047 *

KHFT, knee height feet together; KHVF, knee height involved front; KHNF, knee height normal front; LHFT, low height feet together; LHVF, low height involved front; LHNF, low height normal front. * Significant at α = 0.05.

**Table 6 healthcare-13-00920-t006:** Pairwise comparison results for rectus femoris EMG activity during stand-to-sit at different seat heights and foot positions.

Pairwise Comparisons	*t*	*df*	*p*-Value
KHFT vs. KHVF	1.08	7	0.318
KHFT vs. KHNF	0.31	7	0.769
KHFT vs. LHFT	0.16	7	0.88
KHFT vs. LHVF	0.12	7	0.908
KHFT vs. LHNF	0.14	7	0.896
KHVF vs. NHNF	1.70	7	0.132
KHVF vs. LHFT	5.94	7	0.001 *
KHVF vs. LHVF	2.46	7	0.043 *
KHVF vs. LHNF	2.72	7	0.030 *
KHNF vs. LHFT	0.55	7	0.602
KHNF vs. LHVF	0.55	7	0.598
KHNF vs. LHNF	1.13	7	0.296
LHFT vs. LHVF	0.16	7	0.878
LHFT vs. LHNF	0.90	7	0.399
LHVF vs. LHNF	0.71	7	0.499

KHFT, knee height feet together; KHVF, knee height involved front; KHNF, knee height normal front; LHFT, low height feet together; LHVF, low height involved front; LHNF, low height normal front. * Significant at α = 0.05.

**Table 7 healthcare-13-00920-t007:** Main effect test results for thigh muscle EMG activity during sit-to-stand.

Main Effect	*df*	*F*	*p*-Value
KHFT	2, 6	7.70	0.022 *
KHVF	2, 6	0.44	0.661
KHNF	2, 6	0.007	0.993
LHFT	2, 6	5.72	0.041 *
LHVF	2, 6	3.40	0.103
LHNF	2, 6	3.37	0.104

KHFT, knee height feet together; KHVF, knee height involved front; KHNF, knee height normal front; LHFT, low height feet together; LHVF, low height involved front; LHNF, low height normal front. * Significant at α = 0.05.

**Table 8 healthcare-13-00920-t008:** Pairwise comparisons for EMG activity between thigh muscle during sit-to-stand at KHFT and LHFT.

Condition	Pairwise Comparison	*t*	*df*	*p*-Value
KHFT	VMO vs. RF	1.19	7	0.274
	VMO vs. BF	4.17	7	0.004 *
	RF vs. BF	1.43	7	0.197
LHFT	VMO vs. RF	1.42	7	0.200
	VMO vs. BF	3.64	7	0.008 *
	RF vs. BF	2.12	7	0.071

KHFT, knee height feet together; VMO, vastus medialis oblique; RF, rectus femoris; BF, biceps femoris; LHFT, low height feet together. * Significant at α = 0.05.

## Data Availability

The datasets can be obtained from the corresponding author upon reasonable request.

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
