# Peer review of "Electromyographic Analysis of Thigh Muscle Activity in Arthritic Knees During Sit-to-Stand and Stand-to-Sit Movements: Effects of Seat Height and Foot Position"

_healthcare, 2025, doi:10.3390/healthcare13080920_

Round 1

Reviewer 1 Report

Comments and Suggestions for Authors

Dear Author,

I am honored to submit my review of your article. I think you discussed an interesting subject in a clear way.

The aim of the paper to investigate the activity of the vastus medialis oblique, rectus femoris and biceps femoris muscles during sit-to-stand and stand-to-sit movements in individuals with arthritic knees under varying seat heights and foot positioning conditions has been achieved in a useful way.

The Materials and Methods paragraph is clear and easy to understand, in particular the paragraph Experimental Procedure is well described and reproducible.

The results are presented in a clear and easy to understand way.

The discussion is interesting and offers new perspectives for what concerns different muscles activation, in order to provide patients with more precise and targeted strenghten and rehabilitation protocols to maximize TKA results and minimize patients reported discomforts due to muscular impairements.

The limitations of the study have been clearly exposed at the end of the paper.

Allow me to present some considerations that you might take into account for a major revision:

  • why did the Authors not include the vastus lateralis in the study? Shouldn’t it be more complete by including it?
  • I suggest the Authors to add more patients to their study, since 8 is a very little number. Furthermore, wouldn’t it be more realistic to distinguish between male and female patients, since the osseous and muscular structure may considerably vary between the two genders?

Please justify or correct the two proposed corrections. A minor revision must be performed in order to make the article more valuable.

Reviewer 2 Report

Comments and Suggestions for Authors

This article examines the electromyographic (EMG) activity of three thigh muscles—vastus medialis oblique (VMO), rectus femoris (RF), and biceps femoris (BF)—in individuals with knee osteoarthritis during sit-to-stand and stand-to-sit movements. The study analyzes how seat height and foot positioning influence muscle activation patterns under six different functional conditions. The findings highlight that lower seat heights significantly increase the activation of VMO and RF during sit-to-stand movements, suggesting greater mechanical demand. Foot positioning also affected BF activity, particularly when the arthritic leg was placed behind the non-affected leg.

Major Limitations and Methodological Issues
Syntax error: There is a noticeable syntax issue in line 162: “...each functional movement and condition and are shown...”, which requires correction for clarity.

Missing information: The article lacks a figure illustrating electrode placement, which is essential for reproducibility and understanding the EMG methodology.

Methodological limitation in EMG analysis: The study focuses solely on peak EMG activity, which provides limited insight into muscle coordination or activation timing. A more informative approach would involve analyzing EMG data throughout the entire movement, allowing for a better understanding of muscle synchronization.

Insufficient EMG data extraction: Beyond peak amplitude, the authors could have reported:

Phase-specific activation levels (e.g., early vs. late phase of sit-to-stand)

Motor unit recruitment frequency characteristics, such as mean frequency from a Fourier transform

Small sample size (n=8): This limits the statistical power and generalizability of the findings.

Inconsistent EMG normalization: The normalization method varied between participants—some using maximal voluntary isometric contractions (MVIC) and others using the maximum value during functional tasks, which introduces methodological variability and affects data comparability.

Round 2

Reviewer 2 Report

Comments and Suggestions for Authors

The authors address all the comments made.